# Digital descriptors sharpen classical descriptors, for improving genebank accession management: A case study on *Arachis* spp. and *Phaseolus* spp.

Diego Felipe Conejo-Rodríguez[1,2,3]*, Juan José Gonzalez-Guzman[1], Joaquín Guillermo Ramirez-Gil[4], Peter Wenzl[1], Milan Oldřich Urban[2]*

1 Genetic Resources Program, International Center for Tropical Agriculture (CIAT), Palmira, Valle del Cauca, Colombia, 2 Bean Physiology and Breeding Program, International Center for Tropical Agriculture (CIAT), Palmira, Valle del Cauca, Colombia, 3 Facultad de Ciencias Agropecuarias, Universidad Nacional de Colombia Sede Palmira, Palmira, Valle del Cauca, Colombia, 4 Departamento de Agronomía, Facultad de Ciencias Agrarias, Universidad Nacional de Colombia Sede Bogotá, Bogotá, Colombia

* dfconejoro@unal.edu.co (DFC-R); m.urban@cgiar.org (MOU)

**Data Availability Statement:** The data underlying the results presented in the study are available from https://github.com/agrocompuepidemlab/

## Abstract

High-throughput phenotyping brings new opportunities for detailed genebank accessions characterization based on image-processing techniques and data analysis using machine learning algorithms. Our work proposes to improve the characterization processes of bean and peanut accessions in the CIAT genebank through the identification of phenomic descriptors comparable to classical descriptors including methodology integration into the genebank workflow. To cope with these goals morphometrics and colorimetry traits of 14 bean and 16 forage peanut accessions were determined and compared to the classical International Board for Plant Genetic Resources (IBPGR) descriptors. Descriptors discriminating most accessions were identified using a random forest algorithm. The most-valuable classification descriptors for peanuts were 100-seed weight and days to flowering, and for beans, days to flowering and primary seed color. The combination of phenomic and classical descriptors increased the accuracy of the classification of *Phaseolus* and *Arachis* accessions. Functional diversity indices are recommended to genebank curators to evaluate phenotypic variability to identify accessions with unique traits or identify accessions that represent the greatest phenotypic variation of the species (functional agrobiodiversity collections). The artificial intelligence algorithms are capable of characterizing accessions which reduces costs generated by additional phenotyping. Even though deep analysis of data requires new skills, associating genetic, morphological and ecogeographic diversity is giving us an opportunity to establish unique functional agrobiodiversity collections with new potential traits.

Digital-descriptors-genebank The data of the phenomics and traditional descriptors of the evaluated accessions are associated to this one.

**Funding:** The authors did not receive specific funding for this work.

**Competing interests:** he authors have declared that no competing interests exist.

## Introduction

Characterization of genebank accessions contributes to their use in agro-food systems. However, morphological characterization of large collections is expensive and requires substantial manual labor [1]. CIAT's genebank accessions collection, introduction and first duplications were made in the eighties and nineties. Currently during accession regeneration, the genebank systematically characterizes accessions, collecting basic phenological and morphological data [2, 3]. The genebank attempt to characterize accessions during regeneration. This process has limited outputs mainly by using lower-priority descriptors and because it is still not recognized as important. This is despite the proclaimed need to collect valuable and reliable phenotypic data which could easily be associated with genomic data [4].

Since the late 1980s, plant genetic resources' morphological characterization has been based on the descriptor guidelines from the International Board for Plant Genetic Resources (IBPGR, now part of the Alliance of Bioversity International and CIAT) [5–9]. These classical descriptors have been used to evaluate the phenotypic diversity within the collections and to identify accessions that potentially have characteristics for different scenarios in pre-breeding processes [10]. However, classical descriptors have limitations in germplasm characterization due to the type of data used (qualitative nature which limits genetic association analysis) [11]. High-throughput phenotyping (HTP) methods using proximal sensors and digital images can improve the capture of morphometric and colorimetric descriptors, and monitor physiological traits, and expand the dimensionality of characterization data [12–14]. Phenomic descriptors can complement classical descriptors [15, 16], but the sheer volume, variety and veracity of imaging and remote-sensing data still presents limits in data analysis [17].

Artificial intelligence (AI) promises to deliver real values for humanity including areas such as agriculture and food safety [18]. The high capacity of AI is driven by advances in mechanics, data management, computation, algorithms, and optimization of many mathematics and statistical methods [18, 19]. Machine learning (ML) stands out as a subset of AI as a useful tool for analyzing large datasets generated by non-destructive, field-based, high-throughput phenotyping in plants [17, 20, 21]. However, it has yet to be fully explored to optimize parameters and/or recognize descriptors able to discriminate genebank accessions via HTP analysis [22]. Generating rapid, comprehensive, and precise analysis of large datasets from HTP remains a challenge [17].

Characterization of large genebank collections has been limited due to the high demand of resources and qualified personnel needed even for conventional data capture. Cost-effective phenotyping may involve high initial investments in sensors, vehicles, and method development, but improves quality data capture along with low operational costs [23]. Developing operational, technical, and analytical procedures for genetic resources characterization will contribute to deeper and effective use of germplasm collections. Current characterization procedures do not integrate quantitative metrics for accessions' diversity, distinction, redundancy, and discovery of functional traits. These data are associated with the genome and contribute effectively to managing and documenting germplasm collections. In genebanks, one of the objectives is to conserve a wide range of crop-species diversity based on the components of biological diversity (genetic, taxonomic, ecological, and functional). Unfortunately, diversity-related traits used in genebanks so far do not consider functional diversity evaluated via phenomics as a component to be investigated. Using functional diversity phenomics traits allows understanding the nature and interactions of traits allowing species to flourish within an ecosystem [24]. Therefore, some genebank accessions present functional traits that were directed by processes of natural selection and/or domestication which allowed them to adapt to diverse ecological conditions. In detail, functional diversity indices determine which species exhibit

highly specialized or redundant functional traits and allow increase their resilience when subject to environmental or anthropogenic disturbances [25]. The indices of functional specialization (FSpe), functional identity (FIde) and functional originality (FOri) have been widely used in functional diversity studies in a wide range of ecosystems and organisms [26].

FSpe allows determining the degree to which species exhibit a distinctive set of functional traits compared to species with low FSpe values. These may also contain redundant functional traits. FOri measures the uniqueness of a species' functional traits within a community. A species with high FOri has a unique set of functional traits not found in other species within the same community. FIde measures the functional similarity or dissimilarity among species in a community. The FIde index quantifies to which level a species has similar functional traits with a community and how these traits are distributed among species [25]. These indices have been widely used in studies in diverse ecosystems, however, from the perspective of genebanks they have not been explored.

The genus *Phaseolus*, native to the Americas, is composed of more than eighty species, five of which were domesticated in pre-Columbian times [27]. Beans (*Phaseolus* spp.), and in particular the common bean *P. vulgaris*, represent the most important legume for direct human consumption worldwide [28]. In addition, the forage peanut (*Arachis* spp.) is a native genus of South America, tolerating waterlogging and prolonged drought, and grows well in sun or shade [29]. Forage peanut is mainly used for livestock feeding due to its high protein content and can also be used to improve soil fertility through symbiotic nitrogen fixation [30]. Bean and forage peanut crop wild relatives (CWR) display agronomically interesting traits [31, 32], making them attractive species worldwide and generating increased interest in information associated with CIAT's germplasm collections.

The integration of highly informative digital descriptors, facilitated by HTP and data analysis using AI tools, represents a transformative approach to characterizing genebank accessions. This innovative set of methods should be used together with conventional techniques and augments the efficiency, precision, and informativeness of the outputs. The proposed integration methodology primary objective lies in complementarity with established methodologies and will enhance and optimize the utilization of plant resources by the stakeholders. The proposed synergy between advanced digital tools with traditional methods represents an accurate and insightful paradigm shift of modern plant genetic resources management.

This work propose a methodology to strengthen characterization in genebanks that allows: (1) improving the characterization processes of bean and peanut accessions in the CIAT genebank via identification of phenomic descriptors comparable with classical ones in their discriminatory power; (2) using artificial intelligence (AI) models to identify descriptors that contribute mostly to higher characterization precision and that will quantify functional diversity among accessions; and (3) to integrate the characterization methodology into workflow of genebanks.

## Materials & methods

### Methodological approach

The methodological approach we used is divided into four main phases (for details on used technique, please see below): (1) Taking pictures of flowers, leaves, and seeds at the Palmira and Tenerife seed regeneration stations in the Valle del Cauca. A scanner was used to take leaf images. (2) Preprocessing of images: images are binarized for morphometric analysis and standardized using a color card before being isolated from the background for further colorimetric analysis. (3) Feature extraction of phenomic descriptors: The extraction of RGB color spaces by clustering and morphospaces from contour analysis is done using the R libraries

colordistance and momocx. (4) Classical descriptors: the official list of *Phaseolus* and *Arachis* descriptors was used to determine morphological data, and the RHS color charts were utilized to obtain flower and seed color data. (5) Choosing key descriptors (phenomic, classical, and their combination) for the accessions classification: Random forests were used for the selection process, with the first 10 descriptors being prioritized, starting at the minimal average depth. (6) Functional diversity of accession: The convexhull areas of the 10 functional spaces' accession vertices (functional collection) are identified. (7) The species' functional diversity indicators are shown: The functional entities that join the related accessions will be called species (Fig 1).

## Plant material and cultivation

Scans of leaf samples and flowers, seed and pod photos were taken from Arachis and Phaseolus accessions (S1 *Table*) during the regeneration in the first semester of 2020 at the CIAT Palmira station (03°32′22″N—76°18′13″W; 965 m above sea level; 900 mm annual cumulative precipitation; 60% average relative humidity; 23.8°C average annual temperature) and at the genebank's research station at Tenerife (03°43′34″N—76°04′26″W; 2,200 m above sea level; 1,800 mm annual cumulative precipitation; 70% average relative humidity; 17°C average annual temperature). During regeneration, 16 accessions of Arachis spp. and 14 Phaseolus spp. were selected. Each plot had 120 plants, and 10 healthy and adult plants per plot were chosen to take photos and data. The sample size follows the classical morphological characterization methods [5–7] and three important criteria were used to select accessions for this study: (1) The seed regeneration process: In Tenerife and Palmira, a mapping group of accessions were sown to produce seeds. (2) The present species diversity: In order to support the process of distinguishing and classifying species, we considered the diversity of Phaseolus and Arachis species as well as interspecific hybrids that were present during the particular seed regeneration process. (3) The phenotypic variation: Within a species, accessions were chosen based on morphological, colorimetric variation and seed availability. Since phenotypic characterisation of accessions was fully complementary during the routine seed regeneration process, our study could not change or take an experimental design into consideration.

## Classical descriptors

The descriptors routinely used by genebank during regeneration for forage peanut and beans are based on the IBPGR guidelines for Arachis and Phaseolus species. For *Phaseolus coccineus*, *P. vulgaris*, *P. lunatus*, and *Arachis spp*., we evaluated the following descriptors: days to flowering, days to harvest, weight of 100 seeds, primary seed color, secondary seed color, seed shape, seed color pattern, leaflet shape, standard flower color, flower wing color and pod peak for peanuts [5–7]. The classical descriptors were obtained from historical data from the CIAT regeneration program. The flower and seed color was obtained during photographic captures using Royal Horticultural Society Color card (RHS) which is not (but should be) routinely used in the genebank.

## Image acquisition and pre-processing

Leaf, seed, pod, and flower images were captured for each bean and forage peanut accession. From ten individual plants the ten healthy leaves per accession were selected and images taken. Leaves were then separated into three (beans) or four (forage peanuts) leaflets and scanned using an EPSON 1000XL scanner to generate JPEG files. Flower images were taken in both experimental stations between 10:00 am and 1:00 pm on a sunny day and calibrated via color card. Images of harvested seed and pods were taken with a Canon SX60HS camera in

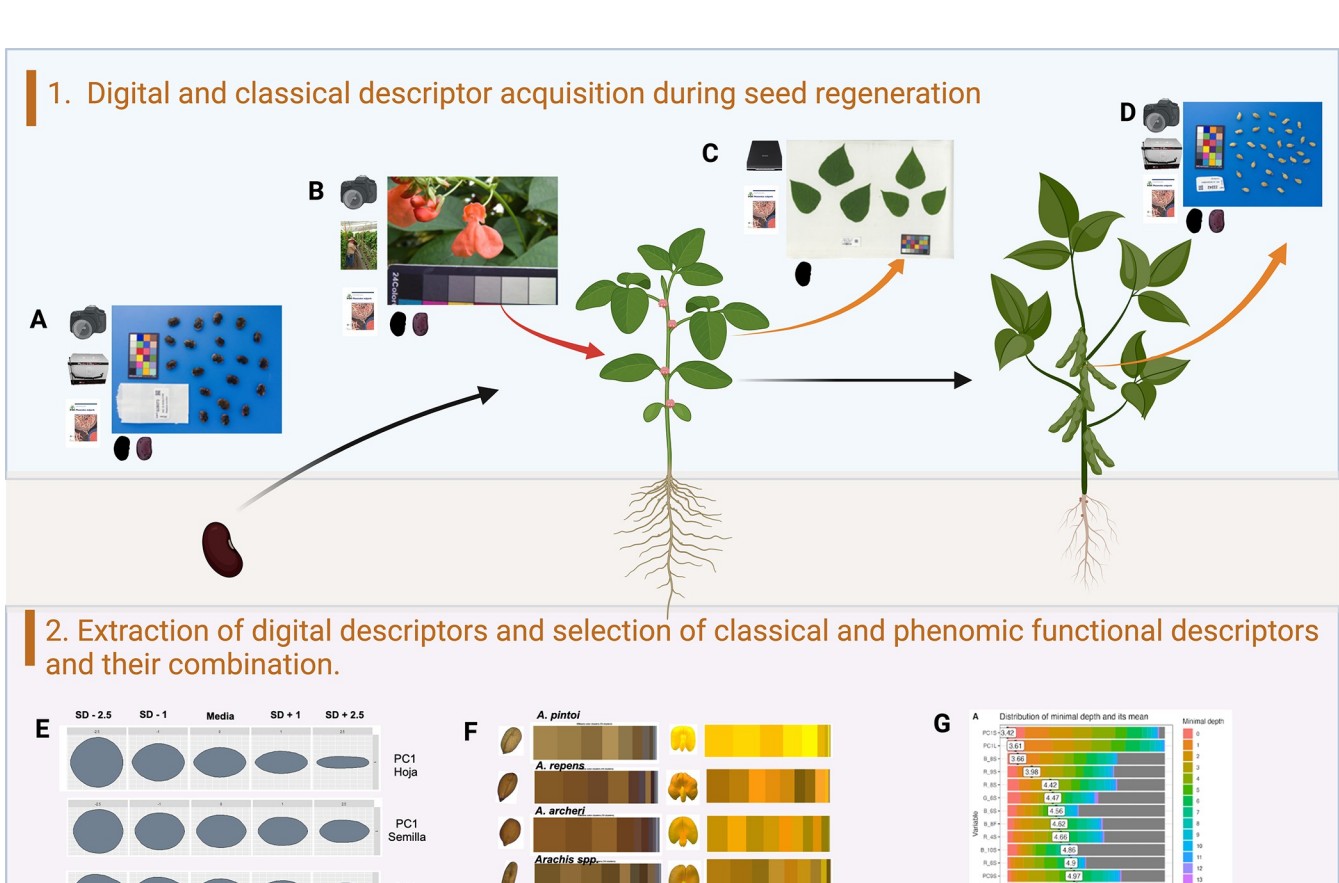

**1. Digital and classical descriptor acquisition during seed regeneration**

**2. Extraction of digital descriptors and selection of classical and phenomic functional descriptors and their combination.**

Contour analysis to seed, leaves and pods - **Morphospaces**

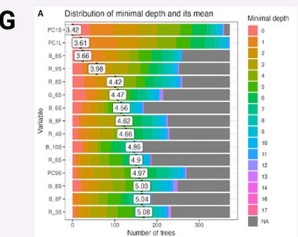

**RGB color spaces** of each cluster in seed and flowers

**Selection functional descriptors** (Classic, Phenomic and combination)

**3. Pipeline data analysis of functional phenome diversity in genebanks**

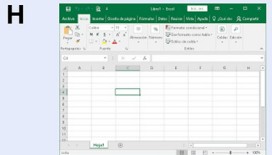

1. Functional descriptors
2. Descriptors class data
3. **Cluster accessions** (Species, ecological and genetic clusters)

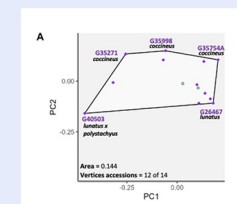

Accessions with greater **variation in functional descriptors** (Vertices of the convexhull)

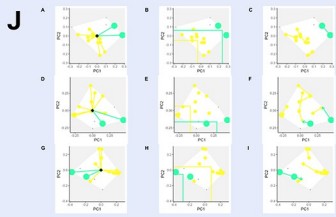

**Functional diversity indexes** among clusters

**4. Icons of processes used during digital phenotyping**

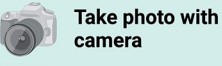 **Take photo with camera**

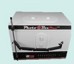 **Photobox**

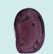 **Single image color**

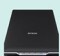 **Take image with scanner**

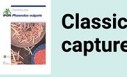 **Classic descriptors capture**

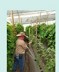 **Take photo in the field**

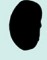 **Single image binarization**

**Fig 1. Methodological procedure for the phenome characterization of *Phaseolus* and *Arachis* accessions.** (1) Digital phenotyping during seed regeneration: A) Seed photographic capture using photobox and classic descriptors. Color and binarize image extraction; B) Field photographic capture of flowers and classic color descriptors. Color image extraction for digital colorimetry; C) Scanning of trifoliate leaves and classical descriptors. Binarized image extraction for geometric morphometrics analysis; D) Image capture of pods in the photo box and classic descriptors. Binarized image is extracted for geometric morphometrics analysis. (2) Extraction of phenomic descriptors and selection: E) Contour analysis using geometric morphometry of seed, leaves and pods. The morphospaces (principal components -PC) are determined; F) Extraction of RGB color spaces from each colorimetric group using kmeans in seeds and flowers; G) Selection of descriptors using random forests from the mean minimum depth. (3) Data analysis of functional phenome: H) Three databases are generated that represent the selected functional descriptors, the class of the descriptors (Character, numerical and ordinal) and the associated groups that are the functional entities (Species, ecological group and genetic group); I) Vertices accessions of the functional spaces. These accessions represent the greatest phenotypic variation; J) Functional diversity indexes that relate accessions to functional entities. Functional specialization (FSpe), functional originality (FOri) and functional independence (FIde) are calculated. (4) Icons relating the stages of digital phenotyping. Image capture and image preprocessing are observed.

ORTech Photo-e-Box Bio to standardize the light exposure [33]. To capture the highest image quality, we used RAW and JPEG formats. A 24ColorCard Camera Trax card was used to standardize the images if taken in somewhat different lighting environments (https://imagejdocu.tudor.lu/plugin/color/chart_white_balance/start). In addition, image backgrounds were extracted for subsequent analyses of color features using the API (Application Programming Interface) PhotoScissors API (https://photoscissors.com/upload). To proceed with the analysis of contours, images were converted into binary format using the ImageJ software with the SIOX plugin [34]. The images were stored and binarized and those used for colorimetric analysis separately.

## Extraction of phenomics descriptors from images

After image binarization of pod, leaflet and seed, the outlines obtained were processed using the Momocs R library to perform a Fourier series analysis. From these we extracted the morphospaces values and the values of classical morphometric descriptors such as length, width, area, and perimeter [35]. The ten (10) main morphospace components from each accession were extracted. For the further analysis, binarized images on the white backgrounds were used to obtain each of the principal components or morphospaces. Each morphospace is considered as a phenomic descriptor that reports the phenotypic variation of Phaseolus spp. and Arachis spp. accessions. The colorimetric analysis was performed using colordistance library R that converts image pixels into three-dimensional coordinates into a defined "color space", producing a multidimensional color histogram for each picture [36]. Pairwise distances between histograms were calculated using the earth-movement distance [37], followed by colorimetric K-means clustering. Ten (10) colorimetric groups were determined and the ratio of each of them is calculated using the R colordistance library.

Each of the colorimetric groups is transformed into a colorimetric descriptor for both seeds and flowers. Each of the RGB values per colorimetric group were used as a phenomic descriptor showing the colorimetric variation within and between accessions (to clarify: R_F1 = cluster Red number 1 of flowers etc.). The list of all used descriptors can be found in the supplementary material (*S2 Table*). Both, the descriptors obtained from the contour analysis and the colorimetric analysis are entered into databases for the selection of descriptors.

## Selection of classical, phenomic and combined descriptors

The random forest (RF) ML algorithm was used to determine the weight of individual descriptors so that evaluated accessions can then be optimally grouped and separated [22, 38]. The RF algorithm was executed using the package caret [39] to create folds (data groups) that are a set of (usually consecutive) records of the dataset and fit the training data set. The classification model used 100 trees and the training was performed with 70% of the data and 30% was used

for validation. The library random Forest [40] was used to run the RF model, and finally, *randomForestExplainer* [41] was used to visualize the RF models.

The relationship between the number of trees and the minimum depth of distribution will determine the most important descriptors able to discriminate accessions comparing the classical, phenomic (digital) and combined descriptors. Out-of-bag (OOB) data accuracy is determined for each descriptor class and its combination presenting values below 25%. This number indicates models of good to high precision in accession discrimination [42, 43]. The highest values of the minimum depth distribution and the lowest number of trees determined fifteen (15) the most important descriptors.

The confusion matrix is made by determining the error proportions of the predicted values for each of the models using classical, phenomic and combined descriptors. The confusion table was analyzed from two perspectives: (1) its high ability to discriminate between non-related accessions/species, and (2) its low sensitivity to confusions between accessions of the same species.

## Functional diversity of phenomic, classical and their combination

In addition to expensive genotype (DNA)-based diversity investigations, where the primary goal is to detect genetic duplicity or redundancy, genebank should also use cheaper phenotypic characterization techniques to confirm if accessions redundant genetically are also phenotypically redundant or not. For this purpose, a functional diversity analysis was carried out aiming to apply the theoretical foundation in the genebank contexts. Initially, each of the most important descriptors was clustered with the related species for both Phaseolus spp. and Arachis spp. and a first matrix "1. *Species—accessions*" was made. Here the presence or absence of each of the accessions grouped in each of the species was considered. Subsequently, a second matrix "2. *Type of descriptor*" was made where the descriptor and its class (number or characters) were associated. Finally, the descriptors selected in the phenomic and classical descriptor types and their combination constituted the third final matrix "3. *Functional descriptors*". Species were treated as functional entities (FE) and accessions as abundances per species. These matrices are required to determine the functional diversity indices. The area of the functional space was calculated for each forage peanut and bean accession descriptor types. Functional diversity indices were determined as: (1) the functional identity (FIde), which has the centroid coordinates of the principal coordinate analysis performed in the functional diversity analysis; (2) the functional specialization (FSpe), which is the average distance from the accessions to the centroid of all species in the principal coordinates analysis; and (3) the functional originality (FOri), which is the average distance between accessions of the same species [44, 45]. These metrics and visualizations were performed using the R mFD library [25]. The metrics were grouped by types of descriptors and functional indices.

Accessions of *P. coccineus* species to *P. lunatus* and *P. dumosus* species, as well as accessions of *A. pintoi* and *A. paraguariensis* species, can be used to visualize the importance of functional indices. We are able to observe the discrimination of every accession for every species for every assessed index thanks to this depiction. Fig 1 shows the methodological process in graphic form.

## Results

### Selection of phenomic, classical, and combined descriptors and accession classification

From image segmentation, processing and analyses, quantitative colorimetric and morphometric values were obtained for bean and forage peanut flowers, leaflets, seeds, and fruits (Figs 2 and 3). Morphometric and colorimetric quantification highlighted wide variation among the

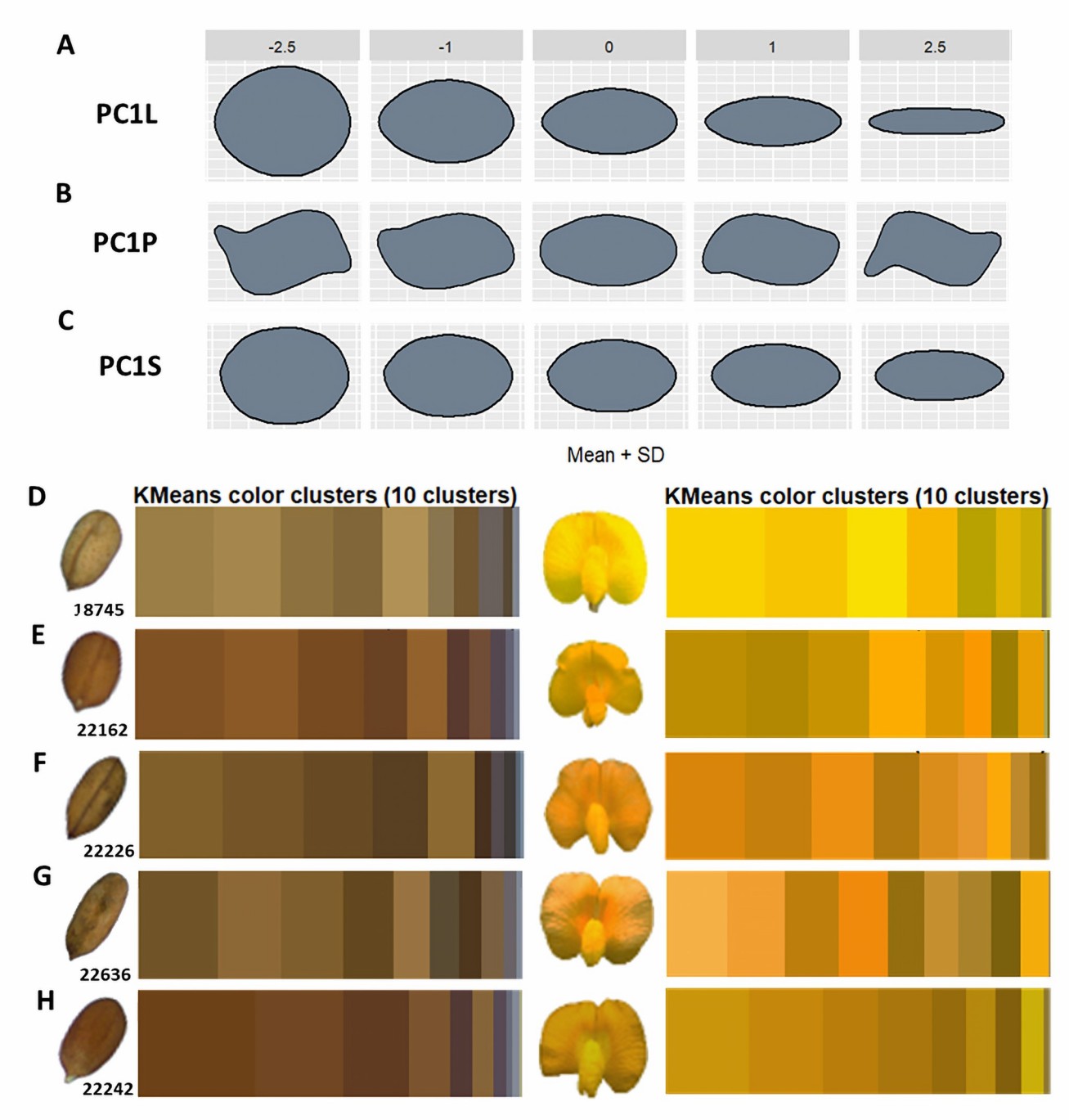

**Fig 2. Functional phenomic descriptors that describe the phenotypic diversity of *Arachis* spp. accessions.** (A) Principal component 1 of leaflet morphometrics. (B) Principal component 1 of pods morphometrics. (C) Principal component 1 of seeds morphometrics. (D) Flower and seed colorimetric clusters of accession 18745 *A. pintoi. (*E) Flower and seed colorimetric clusters of accession 22162 *A. repens.* (F) Flower and seed colorimetric clusters of accession 22226 *A. archeri.* (G) Flower and seed colorimetric clusters of accession 22636 *A. paraguariensis.* (H) Flower and seed colorimetric clusters of accession 22242 *Arachis* sp.

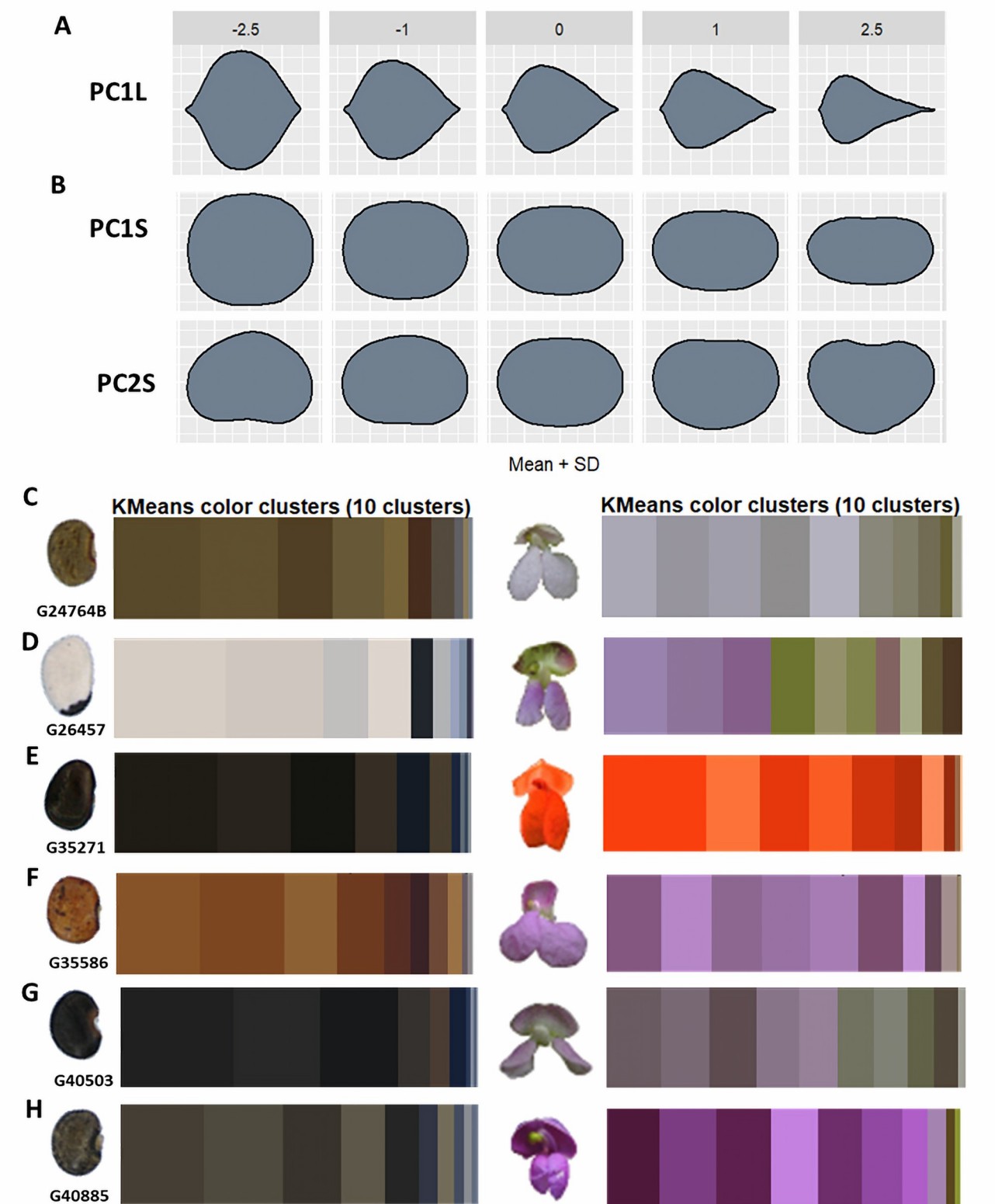

**Fig 3. Functional phenomic descriptors that describe the phenotypic diversity of *Phaseolus* spp. accessions.** (A) Principal component 1 of leaflet morphometrics. B) Principal components 1 and 2 of seeds morphometrics. (C) Flower and seed colorimetric clusters of accession G24764B *P. dumosus x P.*

*vulgaris*. (D) Flower and seed colorimetric clusters of accession G26457 *P. lunatus*. (E) Flower and seed colorimetric clusters of accession G35271 *P. coccineus*. (F) Flower and seed colorimetric clusters of accession G35586 *P. dumosus*. (*G*) Flower and seed colorimetric clusters of accession G40503 *x (P. lunatus x P. polystachyus*. (H) Flower and seed colorimetric clusters of accession G40885 *P. tuerckheimii*.

accessions within both bean and peanut. Based on the random forest algorithm and the minimum depth distribution (MDM) outputs, we selected the most discriminatory descriptors (within the classical descriptors, the phenomic descriptors and their combination) considering the lower the average minimum depth values the descriptor presents, the greater is its interaction with others, and shows greater capacity for accession discrimination. Classification of accessions using the confusion table for each of the descriptors and their combination provides clarity on the descriptor sensitivity in discriminating between individual accessions and their grouping for both bean and forage peanut accessions (Figs 4 and 5).

Variation in pod and seed shape was observed in forage peanuts, determining differences especially in seed and pod length and width and in pod beak shape. In the colorimetry of the peanut flower, there was variation in yellow. In the case of seed we found differences in brown shades (Fig 2). In Arachis, the phenomic descriptors as leaf (PC1LM) and seed (PC1S) and green color of flower (G_8F) are the ones with the lowest MDM values and thus with high variability between accessions. Generally, the descriptors covering the first morphospaces of seed (PC1S and PC5S), leaf (PC1LM and PC3LM) and pod (PC2P) are the most important descriptors in the classification of forage peanut accessions, together with descriptors for seed color (R_10S, G_4S, G_1S, G_3S, B_1S, R_2S, B_6S), pod (R_10P) and flower (G_8F, B_5F) (Fig 4A). In classical forage peanut descriptors, the 100-seed weight (P100_weight), days to flowering (DTF), growth habit and days to harvest (DTH) presented the lowest MDM values (Fig 4C). Other important group of descriptors were those related to the primary and secondary seed color (RHS_sseedcolor; RHS_seedcolor). Similarly, the leaflet shape (Leaflet_shape) descriptor contributes to the discrimination of peanut accessions. The leaflet shape belonging to the classical descriptor showed the same values when taken as the phenomic descriptor (Fig 4C). Importantly, we observed that the Arachis spp. classical descriptors presented a better discriminative power than phenomic ones. For a descriptor combination (classical + phenomic) of Arachis accessions (Fig 4E), the classical descriptors of secondary seed color (RHS_sseedcolor), days to flowering (DTF), 100-seed weight, growth habit and days to harvest (DTH) showed the lowest MDM values and thus highest variability. The descriptors of the first morphospace in pod and leaf (PC1P, PC1L) are good discriminants of forage peanut accessions. Also the RGB color space of seed and flower (G_3S) is very good for peanuts. Generally, compared to the classical descriptors, the phenomic descriptors showed a less number of descriptors suitable for discriminating among forage peanut accessions. In the confusion matrix of phenomic descriptors, *A. pintoi*'s wild accessions (18745, 18746, 1748, 22151, 22159, and 22176) had the highest level of similarity. This indicates that the phenomic descriptors accurately capture the phenotypic redundancy of those accessions where classical descriptors do not show differences. For example, the *A. paraguariensis* (22636) presents similarity with the *A. archeri* (22226) and *A. repens* (22162) indicating phenotypic closeness to these accessions (Fig 4B).

In the case of beans, leaf shape showed high variation among the accessions, allowing differentiation of leaf shape from lanceolate to globular, as well as discriminating seed shape from circular to kidney-shaped (Fig 3). The most important phenomic descriptors in Phaseolus spp. accessions are the first seed morphospace (PC1S) and leaf (PC1L). These two descriptors had the lowest values of MDM, thus high power to discriminate accessions. Seed color descriptors in RGB color spaces (B_8S, R_9S, R_8S, G_6S, B_6S, R_4S, B_10S, R_6S, G_8S and R_3S) in

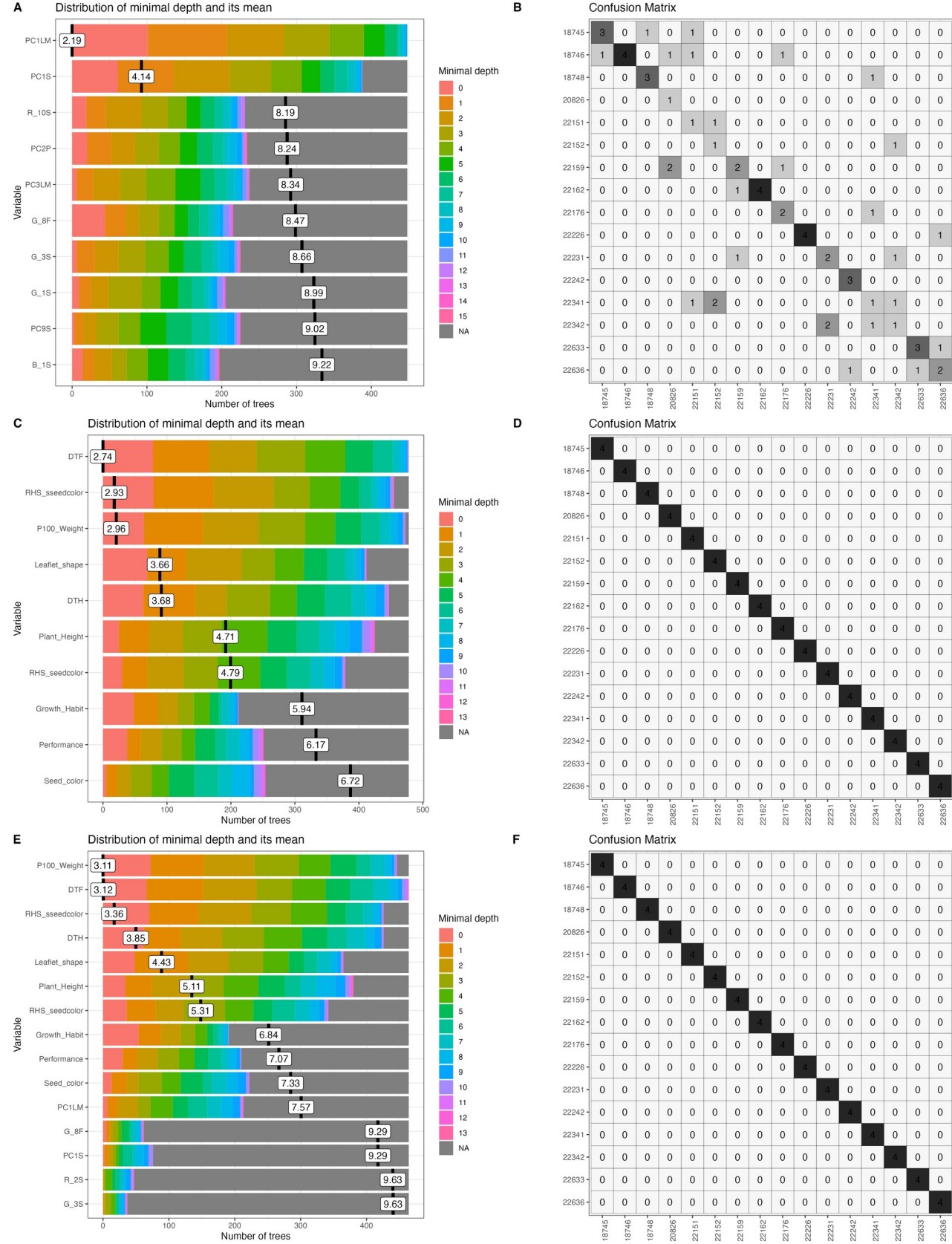

**Fig 4. Selection of classical descriptors, phenomic descriptors and their combination used a minimal depth distribution on *Arachis* spp accessions and its classification.** (A) Distribution of minimal depth of the phenomic descriptors (B) Classification of accessions using confusion matrix from phenomic descriptors. (C) Distribution of minimal depth of classical descriptors. (D) Classification of accessions using confusion matrix from classical descriptors. (E) Distribution of minimal depth of the combination of phenomic and classical. (F) Classification of species using confusion matrix from combining phenomic descriptors with classical descriptors.

addition to the red color space (R) represent highly useful descriptors. This indicates that variability in seed color is the most important trait in discriminating the evaluated Phaseolus accessions (Fig 5A). The days to flowering (DTF), primary seed color (Colorps) and 100-seed weight (Weight100) were the most important classical descriptors in Phaseolus accession discrimination (Fig 5C). Similarly, descriptors associated with seed shape (Formasem), seed brightness and flower color (RHSwing and RHS_standard), leaf length (leaf_length), leaf width (leaf_width) and the ratio between them (ratio_leaf) were highlighted. For descriptor combinations (classical + phenomic), the 100-seed weight, seed shape, primary seed color and days to flowering presented lower MDM values (Fig 5E). Also first seed and leaf morphospace (PC1S, PC1L) and the RGB seed color spaces (B_8S, B_6S, R_4S, R_8S and R_9S) showed better Phaseolus discrimination when combined descriptors were performed (Fig 5E). In the classification of Phaseolus accessions, there is a confusion between accessions of *P. lunatus* (G26467 and G26480), *P. dumosus* and *P. coccineus* (G35580, G35586, G35754A, G36211) (Fig 5A). Interestingly, the accessions that are of interspecific hybridization origin (G24764B and G40503) showed confusion with the species *P. tuerckheimii*, despite being phylogenetically distant (Fig 5B).

For the phenomic descriptors, it is evident that there is a confusion (uncertainty) in both Phaseolus and Arachis accessions. This demonstrates a greater accuracy/precision of phenomic descriptors in detecting redundancy among accessions. Importantly, ML selection of both species (Phaseolus spp. and Arachis spp.) included descriptors of different organs. This indicates a non-biased selection since it counterbalances classical traits used by taxonomists in the classification of plant species with the digital ones. The discrimination of Arachis spp. and Phaseolus spp. accessions using classical and combined descriptors showed no confusion between accessions (Fig 4A and 4F, and Fig 5D and 5F).

## Functional diversity indices are important for beans and forage peanut accessions' characterization

We only considered descriptors prioritized by minimum mean depth (MDM) for each descriptor type. The functional diversity indices are determined to evaluate in quantitative and observable terms the capacity of each descriptor to show phenotypic diversity and redundancy among accessions of the same species. The functional spaces for each of the descriptor types provide evidence of the descriptors' ability to show phenotypic variation among accessions, as well as to determine the descriptors' ability to unravel species. When the functional spaces present a greater area, it is related to a greater phenotypic variation among the accessions.

The diversity indices FSpe, FOri and FIde show the variation between descriptor types and species, allowing grouping accessions that are specialized, or redundant, or distinct (Table 1).

The phenomic descriptors in the bean accessions have a functional space area of 0.144 and 12 accession vertices. The vertices accessions of the first (PC1) and second (PC2) components are accessions G35271, G35998 and G35754A of the species *P. coccineus* and accessions G40503, which is an artificial hybrid between *P. lunatus* and *P. polystachyus* and accession G26467 *P. lunatus* (Fig 6A). In spite of the fact that the classical descriptors showed greater area than phenomic descriptors, they have the same number of vertices accessions (Fig 6C).

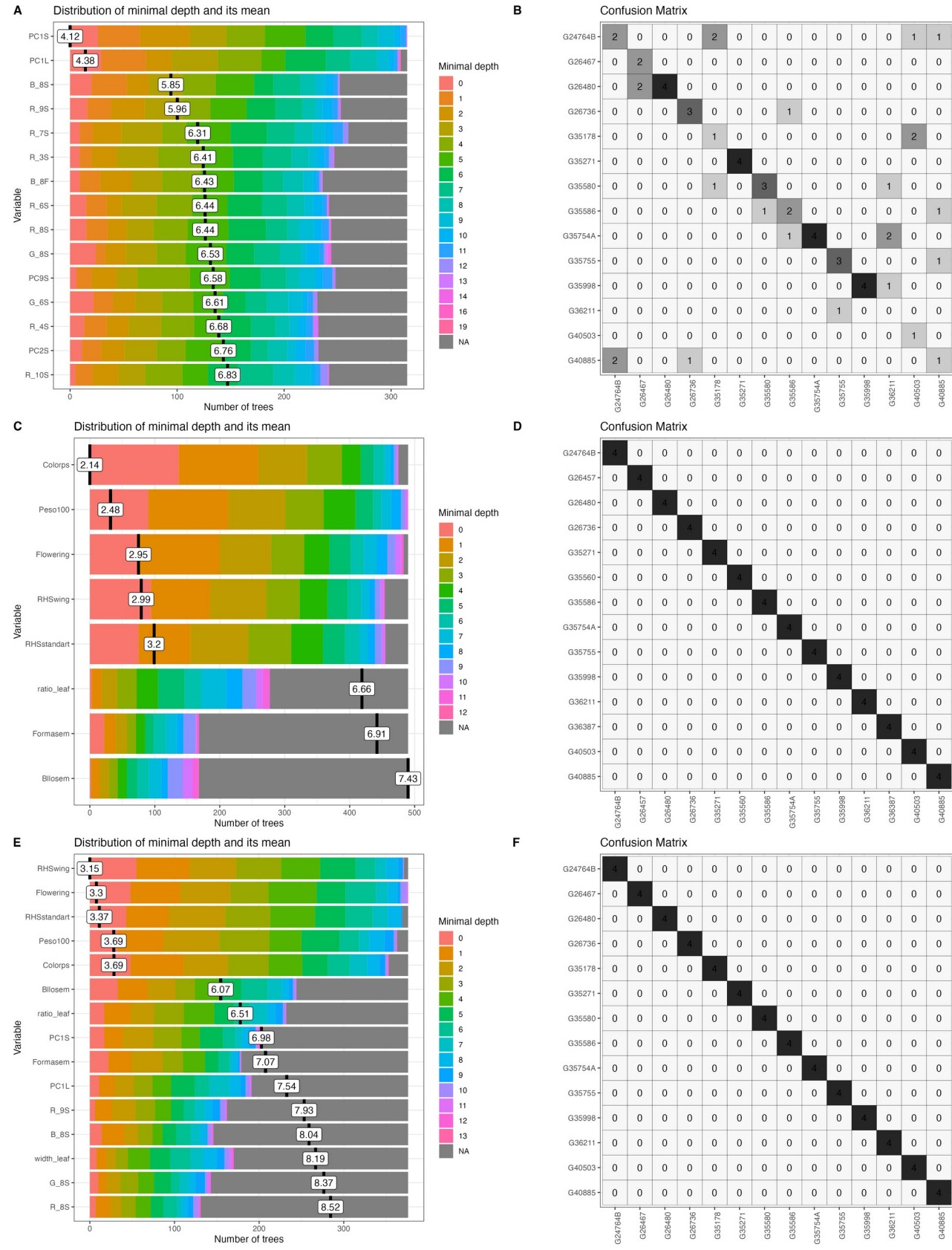

**Fig 5. Selection of classical descriptors, phenomic descriptors and their combination evaluated via a minimal depth distribution on *Phaseolus* spp accessions and its classification.** (A) Distribution of minimal depth of the phenomic descriptors (B) Classification of accessions via confusion matrix from phenomic descriptors. (C) Distribution of minimal depth of classical descriptors. (D) Classification of accessions via confusion matrix from classical descriptors. (E) Distribution of minimal depth of the combination of phenomic and classical descriptors. (F) Classification of species via confusion matrix from combining phenomic descriptors with classical descriptors.

The classical descriptors presented the vertices accessions G40503 (*P. lunatus* x *P. polystachyus*), G35998 (*P. coccineus*), G26736 and G24480 (*P. lunatus*) and the natural hybrid G24764B (*P. dumosus* x *P. vulgaris*) (Fig 6C).

When combined, the phenomic and classical descriptors revealed a larger area in the convexhull than when applied separately. This suggests that combined analysis distinguishes and captures the phenotypic variations of every accession assessed separately. Furthermore, the 14 accessions of Phaseolus (Fig 6E), are displayed as vertices, demonstrating the phenotypic identity of each accession. The functional specialization analysis (FSpe), with the phenomic descriptors, showed a clear discrimination of the species *P. dumosus* with *P. coccineus* (Fig 7A). Calculating FSpe, the classical descriptors do not clearly differentiate between *P. dumosus* and *P. coccineus* (Fig 7G), while when combined the accessions are more separated than using classical descriptors only (Fig 7M). The functional identity analysis (FIde) shows that the phenomic descriptors clearly separate *P. dumosus* and *P. coccineus* (Fig 7C). Importantly, when using only classical descriptors, no clear difference between the two species in PC1 were found (Fig 7I). Finally, the combined descriptors allow a clear distinction between both accessions especially in PC2 (Fig 7O).

When comparing the *P. coccineus* with *P. lunatus*, a clear distinction between them is observed in all three types of descriptors (phenomic, classical and their combination) (Fig 7E, 7K and 7Q). In the combination of descriptors (Fig 7Q), the *P. lunatus* accessions are clearly separated. The separation of *P. lunatus* accessions in the combined descriptors clearly influences the discrimination of accessions. The combination of descriptors in FOri shows a greater separation of *P. lunatus* accessions (Fig 7R) compared to the classical (Fig 7L) and phenomic (Fig 7F) descriptors.

FSpe helps determine that the *P. dumosus* presents its lowest values for phenomic descriptors (0.32) and in descriptor combination (0.47). Meanwhile *P. coccineus* presents the lowest value of 0.45 in the classical descriptors. The *P. lunatus* accessions presented the highest FSpe

**Table 1. Indices of functional diversity for the accessions of Phaseolus and Arachis species evaluated with phenomic (Phe), classic (Cla) and their combined (Comb) descriptors.** The indices of functional specialization (FSpe) and functional originality (FOri).

| Species | Fspe | | | Fori | | |
|---|---|---|---|---|---|---|
| | **Phe** | **Cla** | **Com** | **Phe** | **Cla** | **Com** |
| *P. coccineus n = 4* | 0.45 | 0.40 | 0.62 | 0.51 | 0.33 | 0.60 |
| *P. dumosus n = 4* | 0.32 | 0.61 | 0.47 | 0.47 | 0.25 | 0.26 |
| *P. lunatus n = 3* | 0.67 | 0.82 | 0.84 | 0.78 | 0.62 | 0.73 |
| *P. tuerckheimii n = 1* | 0.23 | 0.88 | 0.55 | 0.37 | 0.52 | 0.25 |
| *x(P.dumosus x P.vulgaris) n = 1* | 0.36 | 0.96 | 0.68 | 0.44 | 0.69 | 0.72 |
| *x(P.lunatus x P.polystachyus) n = 1* | 0.46 | 1.00 | 0.47 | 0.32 | 1.00 | 0.44 |
| *A. archeri n = 1* | 0.78 | 0.82 | 0.86 | 0.76 | 0.60 | 0.85 |
| *A. paraguariensis n = 3* | 0.79 | 0.86 | 0.88 | 0.74 | 0.74 | 0.60 |
| *A. pintoi n = 11* | 0.42 | 0.64 | 0.67 | 0.36 | 0.45 | 0.51 |
| *A. repens n = 1* | 1.00 | 0.68 | 0.76 | 1.00 | 0.75 | 1.00 |
| *Arachis spp. n = 1* | 0.68 | 0.40 | 0.53 | 0.89 | 0.52 | 0.41 |

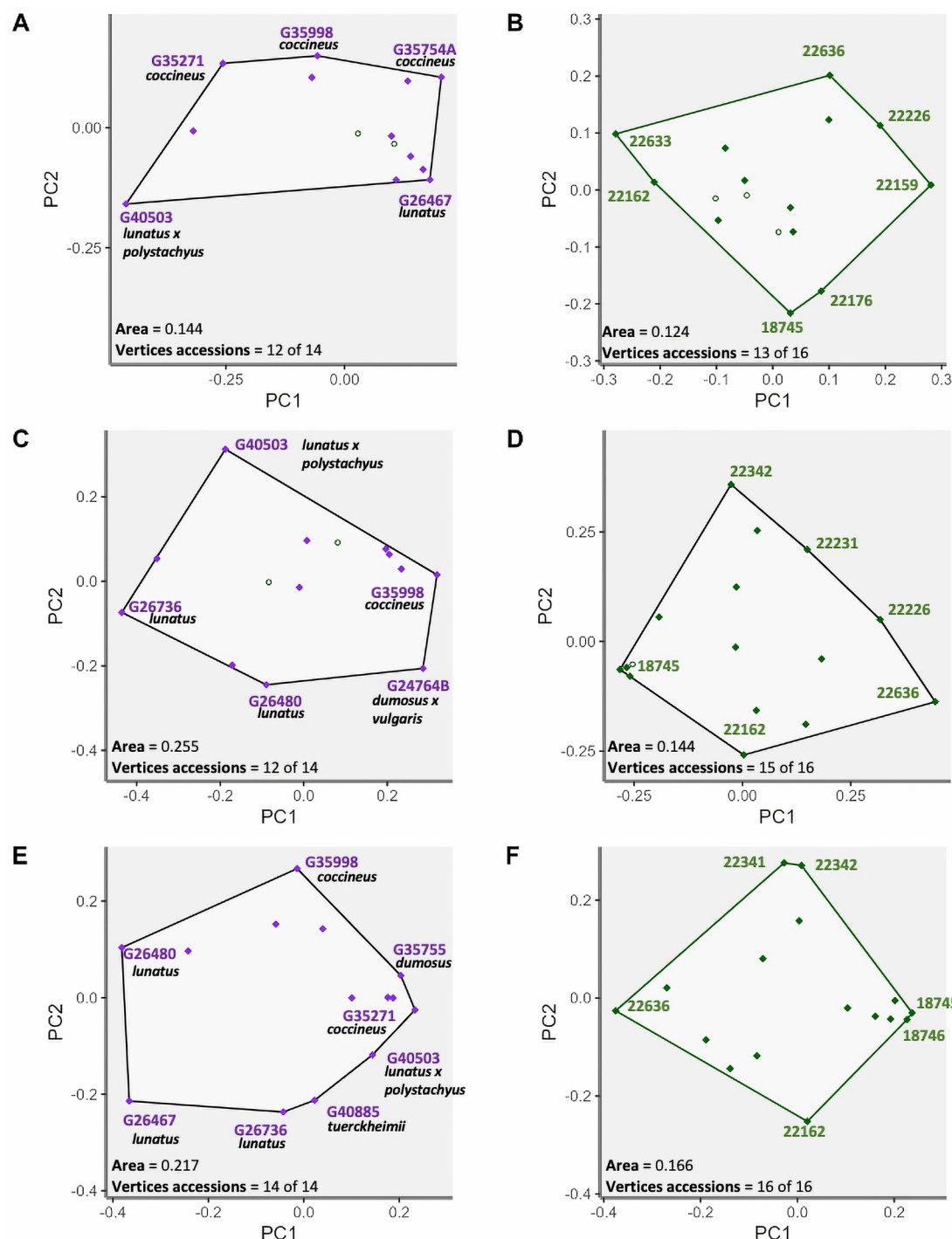

**Fig 6. Vertices accessions of the functional space for each of the descriptor types used for bean and forage peanut accessions.** (A-B) Bean and forage peanut vertices accessions with phenomic descriptors. (C-D) bean and forage peanut vertices accessions with classical descriptors. (E—F) bean and forage peanut vertices accessions with the combination of descriptors. The value of the area of the convex hull and the number of vertex accessions for each of the descriptor types for both beans and forage peanuts.

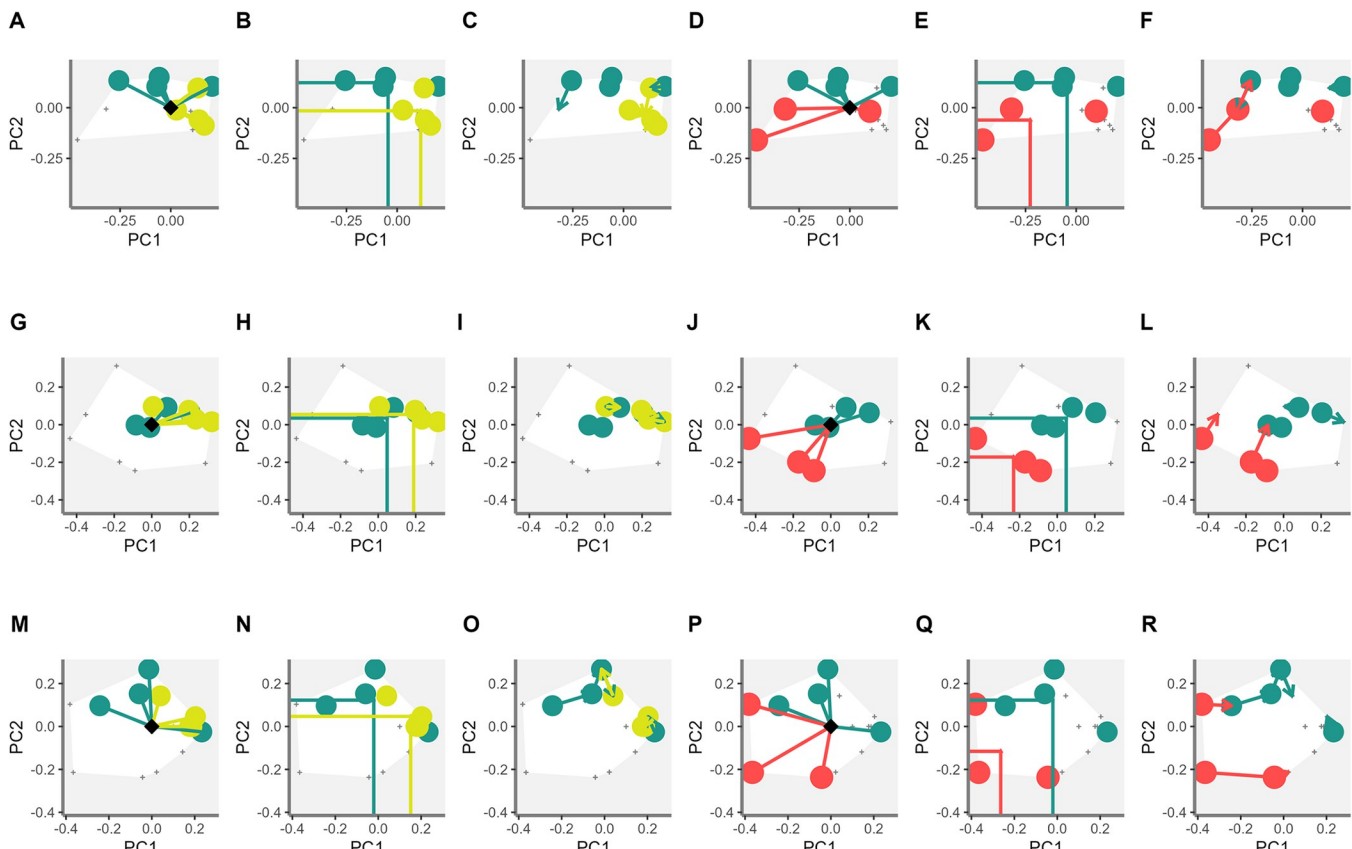

**Fig 7. Functional diversity indices of the accession grouping by Phaseolus spp. for each type of descriptor.** (A-F) Functional diversity indices for phenomic descriptors in the accession grouping of *P. dumosus*, *P. coccineus* and *P. lunatus*. (G-L) functional diversity indices for classical descriptors in the accession grouping of *P. dumosus*, *P. coccineus* and *P. lunatus*. (M-R) functional diversity indices for combined descriptors in the accession grouping of *P. dumosus*, *P. coccineus* and *P. lunatus*. For the indices of functional specialization (FSpe), functional identity (FIde), and functional originality (FOri), the relationship between the first two primary coordinates (PC1, PC2) is observed.

values in the phenomic descriptors (0.67) and the combination (0.84). The artificial hybridization *P. lunatus* x *P. polystachyus* showed the lowest value in the phenomic descriptors (0.32) but the highest value using the classical descriptors (1.00). The FIde index shows that the combination of descriptors generates intermediate values or discriminates to a greater extent (for example *P. lunatus* increasing the distance in the PC1 coordinate; -0.26). The descriptor combination determined that the accession of *P. tuerckheimii* increased discrimination (-0.21), with respect to the phenomic (-0.03) and classical (0.05) descriptors, determining that the accession is a vertices in the convexhull.

In forage peanuts, in two groups of accessions of *A. pintoi* and *A. paraguariensis* were observed the effect of different types of descriptors on functional diversity indices (Fig 8). The FSpe analysis showed the phenomic (Fig 8A) and classical descriptors (Fig 8D) discriminated *A. pintoi* against *A. paraguariensis*. Many *A. pintoi* accessions are clustered using the classical and phenomic descriptors, yet to a greater degree in the combination (Fig 8G). For FIde, the three types of descriptors clearly and similarly separate accessions of *A. pintoi* and *A. paraguariensis* species (Fig 8B, 8E and 8H). In the FOri analysis, the phenomic descriptors allow determining a group of closely related *A. pintoi* accessions (Fig 8C). The accessions of *A. pintoi* are grouped on a cluster of closely related accessions when using the classical descriptors since FOri determines the functional similarity between the accessions (Fig 8F). Importantly, this is

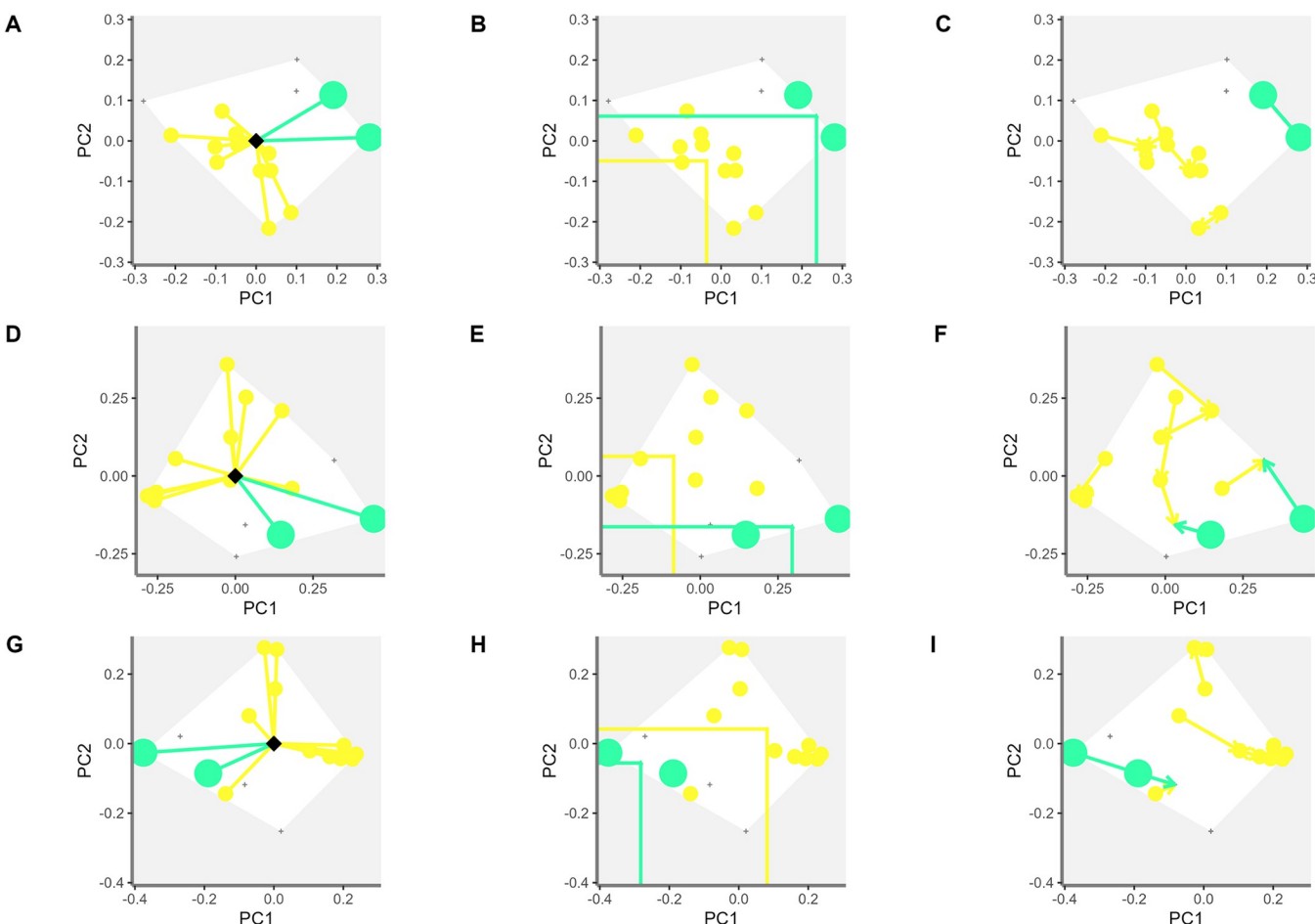

**Fig 8. Functional diversity indices of the accession grouping by Arachis species for each type of descriptor.** (A-C) Functional diversity indices for phenomic descriptors in the accession grouping of *A. pintoi* and *A. paraguariensis*. (D-F) functional diversity indices for classical descriptors in the accession grouping of *A. pintoi* and *A. paraguariensis*. G-I) functional diversity indices for combined descriptors in the accession grouping of *A. pintoi* and *A. paraguariensis*. For the indices of functional specialization (FSpe), functional identity (FIde), and functional originality (FOri), the relationship between the first two primary coordinates (PC1, PC2) is observed.

the confirmation that the descriptor combination method is more sensitive to determine similar accessions (Fig 8I). The FSpe for Arachis determines that the accessions *A. archeri*, *A. paraguariensis* and *A. repens* present greater distance with respect to the centroid. The highest value was observed for *A. paraguariensis* in comparison with the lowest values of accessions of *A. pintoi*. This again demonstrates the high functional redundancy that exists between these accessions.

## Discussion

The results showed that the combination of classical and phenomic descriptors can discriminate between forage peanuts and between bean accessions more accurately than using them independently. For both bean and peanut forage accessions the first morphospaces of seed and leaf (Fig 3A) are important to discriminate accessions. In addition, the variation of flower and seed color spaces were identified as important. This means that the digital quantification of these common classical descriptors as leaf shape and seed color determine the existing morphological variation between the accessions and can speed up the process.

Historically, the variability in morphological and colorimetric traits in leaves, flowers, and seeds was a key factor in determining the genetic representativeness of the collected accessions during the germplasm explorations. List of standardized descriptors relating morphological and agronomic parameters served as a starting point for germplasm introductions into different agro-environments [46, 47].

The classical descriptors as days to flowering, days to harvest, 100-seed weight, and flower and seed colors (Royal Horticulture Society; RHS) allow discriminating both bean and forage peanut accessions (Fig 4C). For example, 100-seed weight is closely related to seed size. That emerged as a result of the domestication syndrome differentiating clearly between wild and domesticated genotypes in modern days. Day to flowering differentiate between different bean species accessions and it is influenced to a greater extent by ecogeographic regions of origin, as was published for both bean and peanut species [27, 48, 49].

Our study—not surprisingly—shows that digital phenomic descriptors have high concordance with classical descriptors. This means that the quantification of leaf shape from geometric morphometry and seed and flower color using both descriptor types will characterize genetic resources with the similar precision [50–53]. The phenomic descriptors such as PC1, PC1L and RGB color spaces for both seed and flower traits thus can be used as descriptors clearly differentiating the variation among accessions and via digital processing save time during and after evaluation.

When we combined both descriptor types, the machine learning analysis selected these descriptors as the most important: PC1L, PC1S, PC1LM, PC1P, days to flowering, PC1P, days to flowering, flower color (RHSwing and RHSstandard), 100 seed weight and classical seed color (RHS_seedcolor) and phenomic (B_6S, R_4S, R_8S, R_9S, G_5S and G_3S) for both bean and peanut accessions evaluated. These descriptors are quantitative, except for the seed and flower color. The above mentioned will allow a better understanding of the accession phenotypic architecture and identify redundant accessions, as demonstrated by the confusion matrices (Fig 5) and the functional diversity analysis (Fig 6).

The phenomic descriptors logically reveal some uncertainty in accessions determination particularly in closely related accessions of forage peanuts and beans. This is not a surprise and will likely change with the newly generated knowledge. However, since all accessions were clearly differentiated, neither the classical descriptors nor the descriptor combination displayed any uncertainty at all. The understanding of the nature of the variables and its overall composition to a species characterization is a key to guarantee the quality of the discrimination, since classical descriptors such as seed brightness (Bllosem) and vigor (performance) are not technically clearly characterized, due to their "*ordinal*" nature. This makes the interpretation difficult in routine operations during seed regeneration processes in genebanks.

Although the classical descriptors presented a better classification than the digital phenomic descriptors, the classical descriptors, being more qualitative, limit its use for genetic studies [54, 55]. Phenomic descriptors aim to digitalize traits and thus quantify them into understandable, determinable, and measurable attributes convergent to genomics. Quantifying and dissecting the responses (physiological, morphological, and other traits) will help pre-breeders, curators, and crop physiologists to understand the complex spatio-temporal dynamics of individual traits orchestrating in different species and then use other techniques (QTL, GWAS, genomic selection etc.) to verify their heritability in different environments and select for them much effectively. This itself should be a valuable justification to add digital phenomics descriptors into any routine characterization of accessions by genebank curators worldwide.

In our study we demonstrate for the first time the use and importance of functional diversity indices in characterizing the genetic resources even possible during the routine regeneration. These indices can become metrics to assess the phenotypic diversity of accessions and

more importantly identify redundancy within collections. One of our key results is the integration of phenomic and classical descriptors.

The descriptors combination allows us to determine a minimum number of accessions, but still represents the highest phenotypic variation in bean and forage peanut accessions (Fig 6) by identifying vertices accessions with differential and/or redundant phenotypic characteristics. Furthermore, the functional space (Fig 6) analysis will allow selection of key accessions in germplasm collections within species.

The conveyesx hull can be evaluated in two ways: (I) Vertices accessions could be considered a core or functional collections; (II) Determining redundant accessions allows genomic evaluations to be performed on a smaller number of accessions and verify their similarity. For example, functional space associates *A. archeri* and *A. paraguariensis* that are closely related to section Erectoides. Also the accessions of *A. pintoi* and *A. repens* under the section Caulorhizae are well associated [56].

In the case of beans, there is a confirmation of the similarity (Fig 6) in functional space of the species *P. dumosus* and *P. coccineus*, as these species share phenotypic and genetic characteristics due to their co-evolution [57]. Importantly, *P. lunatus* is represented by a single group in a functional space showing the discrimination of the combined descriptors between these three species is in correspondence to the recognized gene pools [58]. Functional diversity indices such as functional specialization (FSpe), functional originality (FOri), and functional identity (FIde) [59] have not been implemented in the genebanks so far despite their value. These functional indices have been used to study species in diverse ecosystems and quantifying their level of adaptation to different anthropogenic or natural disturbances [60–62].

One of the main objectives of genebanks is to gather diversity in its different components (taxonomic, ecological, genetic, and functional) through the conservation of accessions that represent the variation in each of the components at both the accession and species levels. This obviously is a very important task of all genebanks, however not easily doable e.g. regarding the missing collections evaluations. There are currently no studies that integrate the characterization of germplasms with metrics associating functional diversity indices.

To understand the importance of our approach by evaluating phenotypic functionality in genebanks, we proposed the following points: (1) Accessions should come from diverse eco-geographic areas (passport data) with determined genetic and phenotypic variation [63], (2) The ability of germplasm accessions to adjust to particular environment is associated with their geographical spread, which is defined by the ecological habitat of origin [64]. (3) The variation of functional phenotypic traits (descriptors) could (in some cases) determine the level of resilience to different biotic and abiotic stress scenarios in the origin [65] but also in the new target population of environments too.

Considering critically the above-mentioned points, FSpe can determine the accessions of each specie that represent extreme functional descriptors, as in the case of *P. lunatus* hybrid (*P. lunatus* x *P. polystachyus;* G40503) [66] (Fig 7) and *A. paraguariensis* (Fig 8). Furthermore, FOri can determine the phenotypic redundancy between accessions per species, as in the case of *A. pintoi*. Species which show the lowest FOri values among the other accessions allow generation of metrics with phenotypic redundancy and thus contribute significantly to the management of germplasm collections.

## Proposed methodology for genebank curators

First, we recommend using the high-throughput phenotyping platforms (HTTP) as an alternative to capture classical descriptor values associated with color, shape, and size of the main plant organs. These can be likely/preferably captured on the plants during the very seed

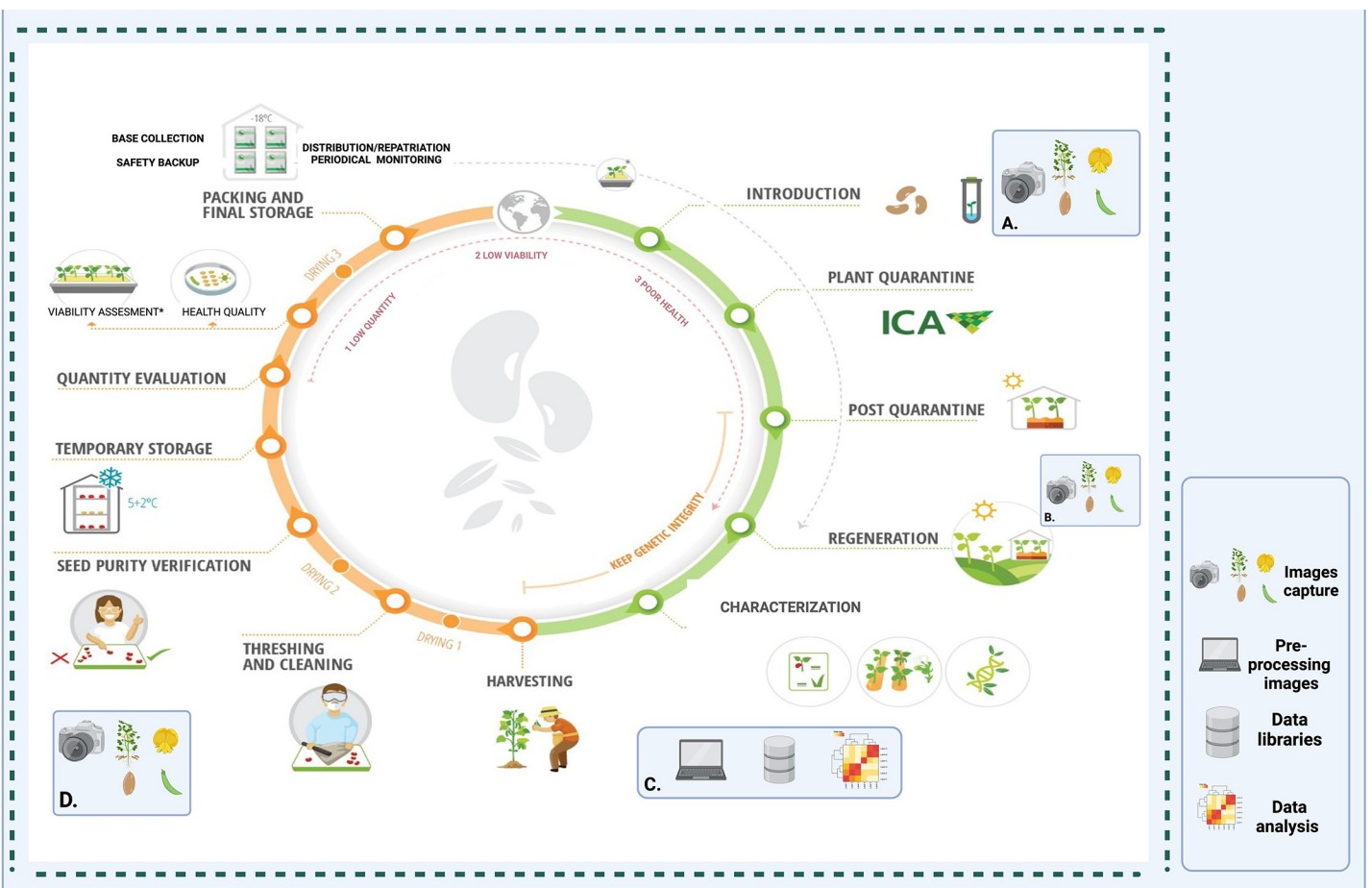

**Fig 9. CIAT genebank workflow suggestion resulting from the proposed methodology.** (A) Capture of images of flower, pods, leaves, whole plant and herbarium sampling and digitization during the collection of new accessions. (B) Capture of images of flower, pods, leaves, whole plant and herbarium sampling and digitization during the seed multiplication and regeneration. (C) Image processing, morphometric and colorimetric data extraction, descriptor selection and data analysis with AI to classify accessions. (D) Capture images of pods and seeds, to feed the database and in the seed purity verification process.

regeneration descriptors into processes in the genebank [50, 52, 53] The workflow of digital image captures must be based on a verified and standardized system by individual genebanks. This guarantees full traceability with the processes of seed regeneration and conservation, as well as required standard operating procedures (SOP) and training [67]. Some of the operational activities from our characterization methodology are depicted in Fig 9. These procedures could be incorporated into the CIAT genebank workflow primarily to facilitate population structure analysis, taxonomic curation, and purity verification. This means that HTTP data needs to be not only successfully captured in the right moment but also transferred in the recognized curator pathways to be added to the database without any confusion. It is not practical to create simultaneous datasets on the same accessions, but mainly to create a very new type of database able to process the classical and phenomic descriptors into meaningful documents. New inventions in HTTP have been giving plant breeders accessible and powerful tools to characterize many genotypes for important agronomic and morphological traits [68] and quantify them. Regarding the traits' names, we strongly recommend that the curated vocabulary is used to describe new digital phenotypic traits and possibly crop trait ontology needs to be applied accordingly.

Second, the image processing performed in different image software alters the results of the tests performed by the artificial intelligence affecting the efficiency of the clustering and discrimination processes [69]. Nowadays with convolutional networks and different trained neural networks it is possible to improve the processes of color correction, image segmentation and feature extraction in an automated way [70]. Descriptors identified by using digital images will allow high efficiency and precision characterization, as demonstrated by our results. The challenge in the image standardization process is to optimize operational processes and follow the technical and infrastructure requirements that could be limited to maintain stable controlled light conditions (quality, quantity, angles) and to have trichromatic sensors of more than 20 megapixels or multispectral sensors [71]. Genebanks also should strategically integrate the use of other genomics and phenomics techniques through the acquisition of high-performance technologies and computational infrastructures (sensors, algorithms, and processing/ storage). That allows efficient and effective accession characterization even during routine genebank operational processes.

Third, analyses of phenotypic data in the genebanks should integrate various tools into one more universal one to reduce data dimensionality, facilitate selection of trait variables and optimize data outputs via machine learning results. The model used for data management will then depend on the objective of study. For example, it will differ for detailed classification of accessions required by curators, in comparison to more general data important for breeders or even deep understanding of accessions adaptability needed by crop physiologists. Our research demonstrates that digital descriptors extracted from organ images like flower and seed, as well as classical descriptors like days to flowering and days to harvest, can be selected to classify bean and peanut accessions. This also means that good curators should be brave and patient but persistent in the process of transformation. Work carried out by others [72, 73], demonstrate the ability to classify species and genotypes of soybean, using machine learning algorithms such as super vector machine (SVM), random forest (RF), neural networks (NN), K-Nearest Neighbors (KNN) and linear discriminant analysis (LDA).

The use of machine learning algorithms is recommended to screen highly characterized training populations. This will contribute to the development of effective automated workflows that could reduce the costs and time currently required for germplasm evaluation and increase the precision of characterization. In addition, using higher spectral resolution proximal and/or remote sensors will certainly contribute to building species-specific data libraries of germplasms (species spectral signatures) to support future digital genebanks.

Fourth, when data capture and selection of functional descriptors is finished, functional diversity analyses can be performed, identifying, and selecting redundant accessions. The evaluation of accessions of interest including functional diversity analyses could be performed in two ways: 1) Evaluation of vertices accessions in pre-breeding processes, integrating selected morphological and physiological traits under stress conditions, especially in crop wild relatives, and 2) Verification of accessions with low FOri values by integrating phenomic and genomic data, identifying probably duplicated accessions in the collection etc. The results of functional diversity analyses will contribute significantly to crop breeding programs, improving and strengthening the parental lines selection and development of new lines that include genes from crop wild relatives that have not been explored for traits associated with tolerance to biotic and abiotic stresses and boost seed nutritional profile (biofortification process). In our previous study we showed that a similar approach can be used to even verify the successful transfer of traits between parents and its progeny [43].

To introduce a new material into a genebank, seed increase is necessary as well as a first characterization of the materials based on classical descriptors. Genetic (genomic) identification can be a significant player in the correct decision-making process too. The regeneration is

carried out when the seed viability or number of conserved seeds has been reduced or for sanitary reasons to eliminate quarantine-type diseases, to conserve the materials in the long term etc. We propose that during these two procedures, trait capturing can be easily performed in the flowering phase (BBCH 65) and then in fruit formation stages (BBCH 75 and on) by capturing photos of leaves, flowers, and pods. If spectral signatures (spectroradiometer) will be taken they need to be stored as digital libraries for later use when calibration curves allow to add additional characteristics. Data needs to be further processed (pre-processing algorithms) and analyzed (machine learning and functional analysis); however, hopefully these processes will be fully automated in the future. In addition, during the seed quality verification (we strongly recommend this step as part of the routine protocol), a parallel path can be established where the capture of seed quality data are associated with seed HQ pictures using various types of sensors (RGB cameras, multispectral, hyper-radiometer) where phenomics descriptors of interest can be extracted and used by stakeholders.

## Genebanks, characterization, phenomics and conceptualization

The genebank collections were collected with the objective of conserving the highest genetic representativeness of wild and cultivated relatives. Both wild and cultivated accessions were in an evolutionary process influenced by environmental conditions and selection pressure during domestication. From this postulate, it can be inferred that the conserved accessions present phenotypic characteristics driven by space (environment), time (evolution) and man (selection) [74]. Based on the above, we could begin to delve deeper into the conceptual aspects that integrate the phenotypic characterization of plant genetic resources and their relationship with phenomics. Germplasm characterization is based on phenotypic traits that have high heritability and can be expressed in all environments [5–7], so that intra-accessional variation is low. Considering the above-mentioned, phenomics is the biological discipline that focuses on the study of phenomes. If the phenome is a set of phenotypes that originate from the relationship between genotype (G), time (t) and environment (E) [75], in the context of plant genetic resources, the accessions collected and conserved today present the phenotypic traits that allowed adaptation to ecological conditions determined by time and environment; therefore, these phenotypic traits are constitutive. The study of the phenotypic variation of germplasm collections (characterization) would thus allow us to evaluate the phenome expression of accessions that are part of the same species and to understand these constitutive functional traits that relate diverse adaptation strategies in the environments in which they evolved. Our work proposes a methodological route that allows the integration of high-throughput phenotyping, artificial intelligence and functional diversity indices that allow exploring the dimensionality of the phenome in genebanks. The study of the phenome in germplasm collections can contribute to the selection of accessions via functional traits with potential for crop improvement and to understand the evolutionary aspects that conditioned the phenotypic diversity of plant genetic resources.

## Conclusions

In our study we demonstrate that via the integration of phenomic and classical descriptors it is possible to identify specific and/or redundant accessions and to characterize each accession (within one species) by using functional diversity indices and AI. Phenotyping via digital images is suitable for germplasm characterization. Image capture can be integrated into various operational processes required during the routine SOPs as seed regeneration, seed quality protocols and seed conservation. The identification of redundant accessions is one of the goals of genebank curators and currently requires the development of new procedures and metrics

in phenotyping characterization. Our proposed methodology facilitates the identification of possible redundant accession groups more easily than the classical approach which is very laborious or almost impossible. Furthermore, we showed that functional diversity analyses using functional diversity indices could be a key in understanding the adaptive capacity of different species. Even though we know the deep analysis of data based on functional diversity indices is not easy and requires new skills, associating genetic, morphological and ecogeographic diversity will establish unique core collections with new potential that have not been explored in genebanks yet.

## Supporting information

**S1 Table. Accession list and passport data.**
(XLSX)

**S2 Table. Phenomic and classical descriptors.**
(XLSX)

## Acknowledgments

We express thanks to Maria Isabel Gomez for her support during image capture in the CIAT entomology laboratory. We would also like to thank the genetic resources program (especially Dr. Marcela Santaella) for the support in methodological development and an overall positive attitude. MOU is thankful to the Gesellschaft für Internationale Zusammenarbeit, Germany (GIZ) for their systematic support. The authors would also like to acknowledge Vincent Johnson of the Bioversity-CIAT Alliance Science Writing Service for his editorial support.

## Author Contributions

**Conceptualization:** Diego Felipe Conejo-Rodríguez, Juan José Gonzalez-Guzman.

**Data curation:** Diego Felipe Conejo-Rodríguez.

**Formal analysis:** Diego Felipe Conejo-Rodríguez, Joaquín Guillermo Ramirez-Gil.

**Investigation:** Diego Felipe Conejo-Rodríguez, Juan José Gonzalez-Guzman.

**Methodology:** Diego Felipe Conejo-Rodríguez, Juan José Gonzalez-Guzman, Joaquín Guillermo Ramirez-Gil.

**Resources:** Peter Wenzl.

**Supervision:** Diego Felipe Conejo-Rodríguez, Joaquín Guillermo Ramirez-Gil, Milan Oldřich Urban.

**Validation:** Diego Felipe Conejo-Rodríguez, Milan Oldřich Urban.

**Visualization:** Diego Felipe Conejo-Rodríguez.

**Writing – original draft:** Diego Felipe Conejo-Rodríguez, Juan José Gonzalez-Guzman, Milan Oldřich Urban.

**Writing – review & editing:** Diego Felipe Conejo-Rodríguez, Joaquín Guillermo Ramirez-Gil, Peter Wenzl, Milan Oldřich Urban.

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
