## [Decision Letter · Decision Letter 0]

10 Oct 2023

PONE-D-23-21318Digital descriptors sharpen classical descriptors, improving the management of genebank accessions: A case study on Arachis spp and Phaseolus sppPLOS ONE

Dear Dr. Conejo,

Thank you for submitting your manuscript to PLOS ONE. After careful consideration, we feel that it has merit but does not fully meet PLOS ONE’s publication criteria as it currently stands. Therefore, we invite you to submit a revised version of the manuscript that addresses the points raised during the review process.

This MS opens new dimension to cope up with digital age. Revision has to be done strictly as suggested by reviewers.

We look forward to receiving your revised manuscript.

Kind regards,

Kuldeep Tripathi

Academic Editor

PLOS ONE

4. Please remove your figures from within your manuscript file, leaving only the individual TIFF/EPS image files, uploaded separately. These will be automatically included in the reviewers’ PDF.

Additional Editor Comments:

Please improve the MS on the recommendation of both reviewers and submit revised form.

Reviewers' comments:

Reviewer's Responses to Questions

**Comments to the Author**

1. Is the manuscript technically sound, and do the data support the conclusions?

Reviewer #1: Yes

Reviewer #2: Yes

2. Has the statistical analysis been performed appropriately and rigorously? 

Reviewer #1: Yes

Reviewer #2: Yes

3. Have the authors made all data underlying the findings in their manuscript fully available?

Reviewer #1: Yes

Reviewer #2: Yes

4. Is the manuscript presented in an intelligible fashion and written in standard English?

Reviewer #1: No

Reviewer #2: Yes

5. Review Comments to the Author

Reviewer #1: The reader has to delve into different parts of the manuscript to understand what the author has presented- for e.g. rationale for selection of material can be worked out only if the reader checks out supplementary data and looks up taxonomic details of species classification. The authors, it is obvious, have a valid basis for selection the species; it has not been presented in the mss.

Reviewer #2: The manuscript “Digital descriptors sharpen classical descriptors, improving the management of genebank accessions: A case study on Arachis spp. and Phaseolus spp.” attractively written and study pertaining to the digital descriptors is new in the field of plant genetic resources for management of genebank accessions.

1. Abstract should be brief and precise elucidating the main content and results of the study

2. Author should include the total germplasm handling of the Arachis spp. and Phaseolus spp in the manuscript available in the genebank.

3. Instead of table, a good quality global map can represent the diverse accessions clearly.

4. There is no discussion about the statistical experimental design for the study.

5. There is vast variability in qualitative traits in Arachis and Phaseolus spp., author include only 5 species, one is unknown (16 accessions) in case of Arachis and 6 species including two Hybrids (14 accessions) in case of Phaseolus. There should be inclusion of more diverse accessions and species (cultivated; Arachis hypogeal and Phaseolus vulgaris) in the study for better representation of qualitative traits to prepare digital descriptors.

6. Sentence in line 169 is not complete.

7. Author is suggested to prepare an informative flow line diagram for the material and methods for better understanding of methodology followed in the study.

The manuscript needs additional information on the aspects cited above, therefore, requires a "Minor revision" for publication of the manuscript in PlosOne to keep up the high reputation of the journal.

6. PLOS authors have the option to publish the peer review history of their article (what does this mean?). If published, this will include your full peer review and any attached files.

Reviewer #1: **Yes: **ER Nayar

Reviewer #2: No

---

## [Author Response · Author response to Decision Letter 0]

6 Mar 2024

I sincerely appreciate your remarks. We'll implement the necessary adjustments. 

I appreciate your suggestions very greatly. We outline the assessed locations that are a part of the International Center for Tropical Agriculture (CIAT) in the methodologies section.

I sincerely appreciate your remarks. I provide a list of the financial support after the question: 

1. The International Center for Tropical Agriculture's bean breeding program provided funding for the technical staff, experts, and materials required for the studies.

2. The genetic resources program, which oversees the genbank, acquired the seed. 

3. The International Center for Tropical Agriculture provided funding for the use of proximal sensors, labs, and computing resources. 

Between CIAT and the Universidad Nacional de Colombia, we collaborate to plan the sampling, analyze the data, choose journals, and prepare submissions. 

We are part of the CIAT and Universidad Nacional de Colombia staff. 

Joaquin Guillermo Ramirez - Associate Professor at the National University of Colombia

Juan Jose Gonzalez - CIAT staff - genetic resources program

Diego Felipe Conejo - CIAT Genetic Resources Program Staff

Milan Urban - Leader of bean physiology in the bean breeding program

4. Please remove your figures from within your manuscript file, leaving only the individual TIFF/EPS image files, uploaded separately. These will be automatically included in the reviewers’ PDF.

I really appreciate your input. We've already fulfilled your request.

I much appreciate your feedback. There are no discrepancies based on our review of the sources. 

5. Review Comments to the Author

We appreciate the reviewers' feedback on the manuscript. The paper submitted to the reviewers includes the responses as an attachment.

---

## [Editor Report · Decision Letter 1]

28 Mar 2024

Digital descriptors sharpen classical descriptors, for improving genebank accession management: A case study on Arachis spp. and Phaseolus spp.

PONE-D-23-21318R1

Dear Dr. Conejo,

We’re pleased to inform you that your manuscript has been judged scientifically suitable for publication and will be formally accepted for publication once it meets all outstanding technical requirements.

Kind regards,

Kuldeep Tripathi

Academic Editor

PLOS ONE

Additional Editor Comments (optional):

Congratulations for novel work.

few last suggestions

1. Use either phenomic descriptors or phenotypic descriptors throughout MS.

2. CIAT name has been changed. Its Alliance of BI & CIAT. so check for name of genebank whther it is CIATs genebank or somethig else. Line no. 63 and 146.